# Anti-Carcinogenic Glucosinolates in Cruciferous Vegetables and Their Antagonistic Effects on Prevention of Cancers

**DOI:** 10.3390/molecules23112983

**Published:** 2018-11-15

**Authors:** Prabhakaran Soundararajan, Jung Sun Kim

**Affiliations:** Genomics Division, Department of Agricultural Bio-Resources, National Institute of Agricultural Sciences, Rural Development Administration, Wansan-gu, Jeonju 54874, Korea; prabhu89@korea.kr

**Keywords:** Brassicaceae, glucosinolates, chemoprotective products, BITC, PEITC, SFN

## Abstract

Glucosinolates (GSL) are naturally occurring β-d-thioglucosides found across the cruciferous vegetables. Core structure formation and side-chain modifications lead to the synthesis of more than 200 types of GSLs in Brassicaceae. Isothiocyanates (ITCs) are chemoprotectives produced as the hydrolyzed product of GSLs by enzyme myrosinase. Benzyl isothiocyanate (BITC), phenethyl isothiocyanate (PEITC) and sulforaphane ([1-isothioyanato-4-(methyl-sulfinyl) butane], SFN) are potential ITCs with efficient therapeutic properties. Beneficial role of BITC, PEITC and SFN was widely studied against various cancers such as breast, brain, blood, bone, colon, gastric, liver, lung, oral, pancreatic, prostate and so forth. Nuclear factor-erythroid 2-related factor-2 (Nrf2) is a key transcription factor limits the tumor progression. Induction of ARE (antioxidant responsive element) and ROS (reactive oxygen species) mediated pathway by Nrf2 controls the activity of nuclear factor-kappaB (NF-κB). NF-κB has a double edged role in the immune system. NF-κB induced during inflammatory is essential for an acute immune process. Meanwhile, hyper activation of NF-κB transcription factors was witnessed in the tumor cells. Antagonistic activity of BITC, PEITC and SFN against cancer was related with the direct/indirect interaction with Nrf2 and NF-κB protein. All three ITCs able to disrupts Nrf2-Keap1 complex and translocate Nrf2 into the nucleus. BITC have the affinity to inhibit the NF-κB than SFN due to the presence of additional benzyl structure. This review will give the overview on chemo preventive of ITCs against several types of cancer cell lines. We have also discussed the molecular interaction(s) of the antagonistic effect of BITC, PEITC and SFN with Nrf2 and NF-κB to prevent cancer.

## 1. Introduction

Cancer is one of the deadliest diseases that cause mortality in humans. According to the report of 2015, cancer killed about 8.7 millions of people [1]. The death rate due to cancer and its-related diseases was expected to be doubled in 2030. Change in the life style of people and environmental pollutions are concerned as the major reason for cancer [2]. Chemo preventive measures gained as a significant strategy for many types of cancers. However, several tumor cells are resistant against drugs. High concentration of chemically synthesized cancer drug brings several side effects such as bruising and bleeding, infection, nausea, short breath, hair loss, fatigue, loss of appetite, diarrhea, skin allergy, infertility and so forth [3]. 

Dietary intake of fruits and vegetables becomes a common practice to avoid the cause of cancers [4]. Plants and its natural products have potential to inhibit cancer [5], reduces the risk of cancer. Several studies previously reported that uptake of Cruciferous vegetables on a daily basis helps to avoid the occurrence of cancer. Glucosinolates are important and economically valuable secondary metabolites mainly found in the Brassicaceae family. Core structure of GSL consists of β-d-thioglucosides *N*-hydroximinosulfates. Subsequently, undergoes various processes such as amino acid elongation, core GSL structure formation and side chain modifications. Structural diversity was gained with addition of methyl groups to the side chain during the amino acid elongation process. Furthermore, molecules undergo repetitive series of transamination, condensation, isomerization and decarboxylation reactions [6]. Core structure formation and side-chain modification lead to the synthesis of more than 200 types of GSLs in Brassicaceae.

Higher accumulation of GSLs is vital for plant defense against biotic and abiotic stresses [7]. Myeloblastosis (MYB) transcription factors (TF) regulate the GSLs biosynthesis [8]. Recently, Seo et al. (2016) found that in *Brassica rapa* (*Br*) genome, *BrMYB28* genes were identified as three orthologous copies to *Arabidopsis thaliana* (*AtMYB28*). Particularly overexpression of three *BrMYB28* genes such as *BrMYB28.1*, *BrMYB28.2* and *BrMYB28.3* are responsible for the higher synthesis of GSL [9]. Subsequently, several activated products of GSLs are beneficial to human and animal health [10]. Myrosinase enzyme catalyzes the hydrolysis process of converting GSL into active substances such as thiocyanates, isothiocyanates (ITCs) and nitrile. Plant derived ITCs are potential chemo preventive agents. Isothiocyanates were characterized as small organic compounds synthesized as glucosinolates with R–N=C=S functional groups. It was converted into active form when the plants were injured or digested. Hydroxylation process was catalyzed by myrosinase or gut bacterial enzymes, respectively. ITCs present in crucifer’s vegetables have higher anti-cancerous property and can inhibit cell proliferation [11]. ITCs suppress the cancer cell proliferation by inhibitions of proteins involved in the tumor initiation and proliferation pathways. Meanwhile, ITCs treatment stimulates the reactive oxygen species (ROS), cell cycle arrest, programmed cell death and autophagy [12]. More than 20 ITCs are reported having anticarcinogenic property against tumorigenesis [13]. Upon consumption of GSLs in the form of Cruciferous vegetables, the presence of myrosinase in human enteric microflora converts the unhydrolyzed GSLs into ITCs. GSLs are more stable and inert whereas ITCs are highly reactive. Metabolize of ITCs are normally taken by the mercapturic acid pathway which rises different dithiocarbamate metabolites [14]. Therefore, GSLs hydrolyzed during indigestion increased the availability of ITCs [15]. Higher ITCs disrupt the several steps of carcinogenesis including the prevention of DNA damage in normal cells, stimulate detoxifying enzymes, cell cycle arrest of cancer cells followed by the induced apoptosis [13]. Allyl isothiocyanates (AITC) is sinigrin derived compound has the potential to cause short-term irreversible DNA damage to the cancer cells [16].

Benzyl isothiocyanate (BITC), phenethyl isothiocyanate (PEITC) and sulforaphane ([1-isothioyanato-4-(methyl-sulfinyl) butane], SFN) are an important ITCs widely studied against various cancer cell lines. Regardless of the origin of cancer cells, BITC, PEITC and SFN inhibit cell growth. Even drug resistant cell lines become sensitive when they are exposed with BITC, PEITC and SFN [13]. Therefore, combination of ITCs with the traditional chemotherapeutic agents also helps to improve the efficacy rate. BITC, PEITC and SFN suppress the tumor growth of various cancer cell lines of breast, brain, blood, bone, colon, gastric, liver, lung, oral, pancreatic, prostate and so forth.

Nuclear factor-erythroid 2-related factor-2 (Nrf2) is an important transcription factor plays a vital role in the cellular defense. Nrf2, a basic leucine zipper (bZip) transcription factor with a “Cap ‘n’ Collar” structure is well demonstrated to play central role in the protection of cells against oxidative and xenobiotic damage. During normal condition, Nrf2 is sequestered by Kelch-like ECH-associated protein 1 (Keap1) [17]. It is constantly ubiquitinated and rapidly degrades Nrf2 through the proteasome pathway. In response to oxidative and electrophilic stress, Nrf2 is released from Nrf2-Keap1 complex and quickly translocate into the nucleus. Higher Nrf2 binds to the *cis*-region of antioxidant responsive element (ARE). Usually, ARE is found in genes encoding for antioxidant and detoxification enzymes. It is vital for the synthesis of GSH, elimination of ROS and drug detoxification and transportation. Therefore, abundance of Nrf2 induced the expression of ARE-dependent genes [18].

As an acute immune response NF-κB directs the cytokines such as tumor necrosis factor (TNF) α, IL-1, IL-6 and IL-8 to the inflammatory sites. However, hyper activation of NF-κB has been contributed for the cancer cell proliferation. Therefore, NF-κB was considered to play “double edged role” in the immune system. On one hand, stimulation of NF-κB directs the leukocytes to the inflammatory sites as part of innate immunity. On the other hand, tumors can principally establish the elevated NF-κB activity by several intrinsic and extrinsic factors. Especially mutation of NF-κB and constant release of cytokines majorly contribute for the cancer progression. In the myeloid cell of the tumor environment, upstream IκB kinase (IKK2)-mediated NF-κB activity contributes to the tumor progression by inducing the secretion of cytokines and growth factors [19]. Study of prostate and breast cancer revealed occurrence of mutation in NF-κB, IKK2 as well as inhibitors IκBα and IκBε [20]. Inflammation in colon cancer is associated with the hyper activation of IKK2-induced NF-κB within intestinal epithelial cells [21]. Cellular location of NF-κB activity is fundamental for the development of liver cancer [22]. In melanoma, IKK2 and NF-κB activation is required for the tumorigenesis [23]. NF-κB activity is mainly involved in the survival of the tumor cells to prevent from the apoptotic or senescent. Therefore, suppression of NF-κB activation is considered as a central mechanism in the prohibition of tumor growth [24]. Therefore, knowledge on the mechanism of NF-κB and its interaction with the beta-catenin/cyclin D1 pathway is essential for finding the novel therapeutic agent(s).

Higher abundance of Nrf2 induces the antioxidant enzymes and its can inhibits NF-κB. An efficient anti-cancer drug should have the ability to control the NF-κB protein complex. Many drugs synthesized to have antagonistic for NF-κB has a pleiotropic effect, that is, compounds able to block NF-κB activation have high probability to inhibits NF-κB’s general physiological roles [25]. In this review, the molecular mechanism(s) of BITC, PEITC, and SFN in inhibition of different types of cancers have been discussed. Role of ITCs on the regulation of NF-κB via ROS-Nrf2-ARE mediated pathway have also discussed elaborately.

## 2. Epidemiological Studies of ITCs

Several epidemiological studies reported that consumption of cruciferous vegetables such as broccoli, cabbage, kale, brussels sprouts and so forth, reduces the incidence of cancer. Broccoli extract can able to induce the phase II detoxification enzymes and the antioxidant enzymes in mammalian cells. It is highly attributed to the presence of SFN. Cooking can substantially reduce activity of ITCs. Compared with the boiled sprouts, uncooked fresh sprouts (with active myrosinase) have more bioavailability of ITCs [26]. ITCs level was increased in the uncooked than the cooked cabbage [27]. Metabolic fate of broccoli after ingestion of steamed or fresh broccoli showed that the bioavailability of ITCs in fresh broccoli is three times higher than the cooked broccoli [28]. Colon cancer mouse model (AOM/DSS) was fed with the different amounts (GSL-poor versus GSL-rich) and different patterns (broccoli versus pak choi) of extract for one month. High glucoraphanin content, broccoli diets induced the formation of SFN–lysine adducts. Amount of 1-methoxyindolyl-3-methyl-histidine derived from neoglucobrassicin was higher in the GSL-rich pak choi group. Though, both GSL-rich broccoli and GSL-rich pak choi up-regulated the expression Nrf2 target genes, colitis and tumor were drastically reduced in GSL-rich pak choi diet [29]. The study in mice with colon cancer showed an anti-inflammatory effect of raw broccoli is higher than the cooked broccoli [30]. Cruciferous vegetable diet displayed significant tumor regression and reduced tumor occurrence in murine models of colorectal carcinoma [31]. Watercress and broccoli extract reduced the colorectal cell proliferation [32]. Intake of cruciferous vegetable significantly reduces the breast cancer risk in the Chinese population [33]. Supplementation of higher soy and cruciferous vegetables to breast cancer survivors received lesser treatment-related to the menopause irregularity and fatigue [34]. Collection of data from past two decades showed that consumption of cruciferous vegetables controls the breast cancer risk factor [35]. Patients with bladder cancer (primary) showed a reduction in tumor proliferation upon the consumption of raw cruciferous vegetables [36]. Bladder cancer can be efficiently prevented by ITCs rich extract [37]. Clinical trial on SFN-extract to the prostate cancer patient decline the cancer risk [38]. In another case study, prostate cancer risk was inverted with the glucosinolate intake [39]. Cruciferous vegetable intake is indeed for the prevention of lung cancer [33]. Cooked cauliflower along with cooked greens modified the risk factor of ovarian cancer [40]. Raw extract from the *B. juncea* induced the apoptosis in several cancer cell lines such as breast (MCF-7 and MDA-MB-231), prostate (PC-3), lung (A-549), cervix (HeLa) and colon (HCT116) cells [41]. Content of GSLs can be varied in stored vegetables also processing could degrade the active compounds [6]. Dietary intake of SFN in the form of BroccoMax^TM^ to the ten healthy dogs was peaked after 4 h from plasma concentration of SFN and SFN metabolites. In the canine patients uptake of SFN could have inhibited histone deacetylase (HDAC) activity [42]. It is also debatable that as GSLs hydrolyzed by gastrointestinal microbes, consumption of broccoli significantly reduced the gastrointestinal microbiota [43]. Nevertheless, epidemiological studies on human and animal proved that uptake of cruciferous vegetable could reduce cancer risk. It is also showed that heat inactivates the myrosinase activity and subsequently reduced the bioavailability of ITCs. Therefore, to reduce the cancer risk factors either fresh vegetable or raw extract consumption is more efficient than the cooked or boiled one.

## 3. Molecular Mechanism of ITCs in Cancer Prevention

Each ITCs treatment has the potential targets on cell signaling (inhibit or enhance) for the prevention of cancer cell growth. Here we discussed BITC, PEITC and SFN effects in different cancer cell lines.

### 3.1. Benzyl Isothiocyanate

Benzyl isothiocyanate (BITC) is a hydrolyzed product of glucotropaeolin in cruciferous vegetables. BITC is highly abundant in cabbage, garden cress and Indian cress [44]. It has antioxidant, anticancer and antimetastatic properties. BITC has been found to exhibit the prevention of various cancers (Table 1).

#### 3.1.1. Blood

BITC prevent the blood cancer cell proliferation by blocking the extracellular signal-regulated kinases (ERK)1/2 and c-jun *N*-terminal kinase (JNK) preceded cyclooxygenase-2 (COX-2) suppression [45]. Lee et al. (2017) showed that ITCs inhibit the C6 giloma cells by downregulating the focal adhesion kinase (FAK)/JNK-mediated matrix metallopeptidase-9 (MMP-9) expression [46]. Tang et al. (2015) showed that BITC treatment inhibits the proliferation of blood cancer cells was correlated with a decrease in the expression of mitogen-activated protein kinase (MAPK) and nuclear transcription factors (TFs) [47]. ERK and MAPK signaling pathway play a vital role in the cellular differentiation. Malfunction in the ERK and MAPK pathway stimulate growth factors, cytokines and carcinogens signals. Especially higher activation of Ras, growth regulator activates the uncontrollable cell growth [48]. Ability of BITC to regulate the ERK1/2 and MAPK prevent the cancer cell proliferation. COX-1 is a “housekeeping genes”. Higher expression of COX-2 is the biomarker of inflammation [49]. MMP-9 involved in the extracellular matrix remodeling and angiogenesis [50]. An abundance of MMP-9 facilitates tumor progression and invasion. BITC decrease the expression of MMP-9 by blocking the nuclear translocation of NF-κB and activator protein (AP-1). Both NF-κB and AP-1 linked with the cytokine and the cell survival mechanisms. It induces the apoptosis of C6 blood cancer cell lines [51]. Detailed functional role of NF-κB and interaction of BITC will be discussed in Section 4.

#### 3.1.2. Breast

Women with breast cancer suffer more pain due to osteolytic bone resorption. Though, BITC inhibits breast cancer, lesser expression of NF-κB and runt-related transcription factor 2 (RUNX2) induce osteolytic bone resorption [52]. RUNX2 transcription factor is a core-binding factor subunit alpha-1 (CBF-alpha-1) for osteoblast differentiation [53]. Activation of p53-Liver kinase B1 (LKB1) and p73-LKB1 prevent the growth of several breast cancer cell lines in BITC treatment [54]. During the cancer cell proliferation, tumor suppresses the p53 (either silenced or mutated). However, p53 expression is necessary to suppress cancer cell growth, angiogenesis and migration of cancer cells. The p73 is a TF with more structural and functional homology with p53 [55]. LKB1 is involved in the tumor-suppressor by regulating downstream pathways. Apoptosis and autophagy of breast cancer cells stimulated by BITC were mediated with upregulation of forkhead box (FOX) protein H1 (FOXH1) pathway. Higher expression of FOX promotes the tumor metastasis and invasion [56]. Breast tumor cell death was associated with acetylation of FOXO1 in rhabdomyosarcoma under BITC treatment [57]. FOXO1 acetylation leads to the autophagy cell death by the induction of H_2_O_2_ and serum starvation. Environmental stress signals stimulate the transcriptional activity of FOXO1. Importantly, FOXO1 TF involved in the tumor suppression and other developmental process by modulating protein kinase B (AKT)/PKB pathway [58].

#### 3.1.3. Brain

Inhibition of brain tumor is involved with the down regulation of protein kinase C ζ (PKC). PKC plays as major role in the cell signaling and inflammation. Arrest in G2/M cell phase of U87MG cell was also associated with suppression of cyclin B1, p21, MMP-2/9, vascular endothelial cadherin (VE-cadherin), C-X-C chemokine receptor type 4 (CXCR4) and MutT homolog 1 (MTH) [59]. Especially, MTH regulate the DNA repair of cancer cells is not required for normal cells. BITC targets the MTH to cause the DNA damage of tumors. Tang et al. (2016) reported that BITC helps to prevent brain tumor involved with the suppression of tumor protein p52, mitochondria ribosomal protein (MRP) S2 and MRP L23 [60]. The p52 NF-κB subunit is a basic regulator of the gene associated with chromosomal translocations of B- and T-cell lymphomas. Higher activation of cysteine aspartic acid protease (caspase)-3 or CASP-3 and BCL2 associated X (bax) found in BITC treatment in brain tumor cell stimulate the apoptosis [61]. Induced bax expression translocate the mitochondrial membrane promotes CASP3 activities to trigger the apoptosis. 

#### 3.1.4. Colon

Interestingly, depletion of cholesterol in BITC treatment inhibits the phosphatidylinositol-3-kinases (PI3K)/AKT dependent survival pathway of colon cancer cell lines [62]. Meanwhile, BITC simultaneously up-regulated apoptosis related proteins and down-regulated metastasis-related proteins [63]. In most of the cancer cells, aberrations of PI3K/AKT/mTOR pathway are the common abnormalities. Blocking the pathway of PI3K and AKT directly inhibit the cancer cell proliferation [64]. This pathway is involved in glucose metabolism, apoptosis, cell proliferation and migration. Inflammation of colon cancer was reduced under BITC treatment. Antimetastatic role of BITC was demonstrated in HT29 colon cancer cells. Metastasis process of the tumor was altered with the suppression in the expression of MMP-2, MMP-9 and urokinase-plasminogen activator (u-PA). Further reduction in the phosphorylation of JNK1/2, ERK1/2, PI3K and PKC inhibit the DNA binding activity of NF-κB [65]. The migration of colon cancer cells was slow down with BITC treatment.

#### 3.1.5. Pancreatic

BITC inhibit the phosphorylation of critical amino acids of AKT. This results in the arrest of PI3K/AKT pathway. Prevention of AKT phosphorylation also decreased FOXO1 and FOXO3a in pancreatic cancer [66]. Inhibition was mediated by the downregulation of VEGF and the cluster of differentiation 31 (CD31). VEGF is a signal protein stimulates the formation of new blood vessels. In cancer cells, VEGF overexpression is a main step for the adequate blood supplies to stimulate the abnormal growth and metastasize. Lower expression of VEGF correlated with the lesser abundance of CD31, a glycoprotein majorly present in platelets, endothelial cells and leukocytes [67]. Excessive generation of ROS cause the DNA damage as well as stimulate the H2A histone family member X (H2A.X) and checkpoint kinase 2 (Chk2). Important amino acids involved in the activation of histone H2A.X and Chk2 was Ser139 and Thr68, respectively [68]. Higher expression of H2A.X and Chk2 induce the apoptosis, tolerance against DNA damage and DNA repair. BITC treatment sensitizes pancreatic cells to TNF-related apoptosis-inducing ligand (TRAIL) for tumors cell death [69]. TRAIL is a cytokine and produced in normal tissue. It can able to stimulate the apoptosis of tumor cell by interaction with the death receptor, TNF receptor superfamily (TNFRSF). Antineoplastic activity of BITC was correlated with its signal transducer and activation of transcription 3 (STAT3) pathways. Increase in the expression of STAT3 controls the conformational change of inflammatory-related proteins. Additionally, BITC target and decrease the expression of specificity protein (Sp), STAT3 [70]. Mechanisms of BITC to control cancer cell growth were described in Figure 1.

### 3.2. Phenethyl Isothiocyanate

Though phenethyl isothiocyanate (PEITC) is found across crucifers, it is highly abundant in water cress (Table 1) [44]. Precursor of PEITC is gluconasturtiin. It has a higher potential to induce ROS and oxidative damage to cancer cells.

#### 3.2.1. Blood

PEITC reduced the growth of blood tumor cell by inducing Fas and Fas ligand (FasL) expression and ROS generation [71]. Fas is a member of the death receptor family (type I membrane protein) and FasL is type II transmembrane protein, belongs to TNF family. Fas-FasL binding plays a fundamental role in apoptosis [72]. This interaction triggers the programmed cell death as an immune response. Subsequent release of cytochrome c and stimulation of caspase-3 and -9 caused the cell death [71]. Chronic lymphocytic leukemia (CLL), affects mainly the adult and it is also resistant to fludarabine (chemotherapy medicine used to treat leukemia and lymphoma). Exposure of both resistant and sensitive CLL cell lines to PEITC leads to massive cell death due to the glutathione depletion, higher ROS generation and oxidation of mitochondrial cardiolipin [73]. Similarly, in another study on chronic myelogenous leukemia (CML), cancer on white blood cells is resistant to Gleevec due to the mutation on Bcr-Abl. Apoptosis of T3151 CML cells (wild- and *Bcr*-*Abl*-mutant) in PEITC treatment was related to elevated level of ROS and ROS-mediated degradation of Bcr-Abl protein [74].

#### 3.2.2. Breast

Reduction of human epidermal growth factor receptor 2 (HER2) and STAT3 promote caspase-3 and poly(ADP-ribose)polymerase (PARP) in breast cancer cell lines [75]. HER2 is a member of the epidermal growth factor receptor (EGFR) family. Higher expression of HER2 was witnessed in many types of cancers. HER2 regulate cell growth, survival and differentiation. Excessive cell growth can be controlled by decreasing the expression of HER2 [76]. ITCs stimulate the DNA fragmentation and PARP cleavage. PARP involved in the DNA repair, genomic stability and programmed cell death. Usually, PARP is activated in the cell under the stress condition. Higher abundance of PARP leads to the ATP depletion during the repair process of damaged DNA. Consequently, lower level of ATP leads to cell death [77]. Apoptotic process was associated with the higher expression of caspase-3 and -9 followed by the ROS generation [78]. Hyperactivation of PARP involved with NAD^+^/ATP depletion is a potential target of cancer therapy [77]. Combination of PEITC with taxol (commercial drug) effectively inhibited the breast cancer cell lines than alone. Cell proliferation was prevented with reduction in the expression of cyclin-dependent kinase 1 (cdk1) (cell cycle regulator) and B-cell lymphoma 2 (bcl 2) (anti-apoptotic protein) [79]. Cdk1 involved in the many processes of cell division such as G1/S phase transition, DNA replication in S phase, nuclear breakdown, chromosome condensation and segregation and cytokinesis. Family members in bcl are also involved in the anti-apoptosis. Some of bcl-2s are also involved with the pro-apoptosis. Immune response against breast cancer in PEITC treatment was correlated with level of CD19 [80]. CD19 is an antigen of B-lymphocyte. PEITC interfere with angiogenesis via preventing the HIF family protein [21]. Proteins belonging to the HIF promote the formation of blood vessels. PEITC is an efficient HIF inhibitor to stop the angiogenesis of tumor cells.

#### 3.2.3. Colon

Similar to BITC, PEITC also possesses antimetastatic property. Invasion and migration of colon cancer was prevented by the inhibition of genes involved in the cell cycle such as the son of sevenless homolog 1 (SOS-1), PKC, ERK1/2 and Ras homolog gene family, member A (Rho A). SOS is critical for cell growth and differentiation. RhoA associated with the cytoskeleton regulation directly alter the cell signaling proteins for inflammation or cell growth, MMP-2, MMP-9, Ras, focal adhesion kinase (FAK), PI3K, growth factor receptor-bound protein 2 (GRB2), NF-κB, inducible isoform of nitric oxide synthases (iNOS), COX-2, AKT and JNK [81]. Inflammation of colon cancer was reduced under PEITC treatment. Induction of mitochondria caspase cascade and JNK, critical factors for the initiation of apoptotic was observed in PEITC treated HT-29, colon cancer cell [82]. Anti-inflammatory effect of PEITC against colon cancer was correlated with the lesser expression of NF-κB [83].

#### 3.2.4. Ovary

Ovarian tumor cells treated with PEITC showed lesser phosphorylation rate of EGFR/AKT pathway. Blocking the upstream region of Akt such as EGFR and HER2 prevents the proliferation of cancer cells. Blocking the HDAC inhibit the androgen receptor in prostate cancer. HDAC is associated with cell proliferation, cell cycle regulation and apoptosis which are linked to the tumor development. Oncogenic pathways such as EGFR, HER2 and Akt were inhibited by the PEITC [84]. PEITC decreased expression of chromosomal maintenance 1 (CRM1) proteins [85]. CRM1 is required for the centrosome duplication and spindle assembly. It can facilitate the transport of large macromolecules across the membrane. Ovarian cancers cell resistant to cisplatin (commercial drug) turned to sensitive after treatment with the metformin and PEITC. Higher ROS slower down the growth and subsequently kill the ovarian cancer cells [86].

#### 3.2.5. Prostate

For preventing pancreatic cancer, PEITC upregulates the 8-oxo-deoxy guanine and pH2AX foci [87]. Induction of ferroptosis inhibits the proliferation of pancreatic cancer cell lines [88]. Exposure of prostate cancer cells to the PEITC reduces suppression of Bcl-2 and X-linked inhibitor of apoptosis protein (XIAP) levels and induction of Bax and Bak. It disrupts the mitochondrial electron transport chain as well as reduces oxygen consumption [89]. In another prostate cancer study, translocation of p66^Shc^ to the mitochondria on PEITC treatment induced the apoptosis and suppressed the tumor growth [90]. Inhibition of VEGF and HIF-1α reduces the angiogenesis of cancer [91]. Higher autophagy observed in cancer cells treated with PEITC was associated with the ARE-mediated pathway [21]. Upregulation of *hexokinase II* and *lactate dehydrogenase A* suppresses glycolysis of prostate cancer cell lines [92]. Reactivation of mutated p53 also inhibits the cell proliferation and induces apoptosis in prostate cancer cell lines [93]. Exposure of prostate cancer cells to PEITC downregulate many fatty acid metabolism proteins, including acetyl-CoA carboxylase 1 (ACC1), fatty acid synthase (FASN) and carnitine palmitoyltransferase 1A (CPT1A) [94]. For treatment of glioma, PEITC regulates the MRP1-mediated export of GSH to activate ROS-MiR-135a-Mitochondria dependent apoptosis pathway [95]. G2/M phase arrest induces mitochondrial apoptosis. This process is mediated by the increase of p53 and WEE1 with the reduction in the level of cell division cycle (CDC) 25C protein, a subclass of protein tyrosine phosphatases. CDC25 proteins actively involved with the various phases of cell cycle. WEE1, a kinase controls the cell growth and inhibit of cdk1. Higher expression of WEE1 constrains the mitosis of cancer cells. Following, the induction of apoptosis was supported with the activation of caspase-3, -8 and -9 dependent pathways of prostate cancer cells [96]. Metabolic pathways involve by PEITC to prevent cancer was given in Figure 2.

### 3.3. Sulforaphane

Sulforaphane (SFN) (1-isothiocyanato-(4*R*)-(methylsulfinyl) butane: CH_3_S(O)(CH_2_)_4_–N=C=S), a naturally occurring cancer chemopreventive agent found with precursor glucoraphanin (Table 1). SFN is the most potent inducer of Nrf2 and presences of oxygen on sulfur enhance its efficiency. Broccoli, brussels sprouts and cabbage have higher amount of SFN [44].

#### 3.3.1. Brain

Treatment of SFN to the brain tumor cells induced the translocation of Nrf2 to the nucleus [97]. Activation of ERK1/2 and downregulation of MMP-2 and CD44v6 inhibit the brain tumor invasion [98]. For the migration, transformation, invasion and generation of metastatic tumors, CD44v6 is an essential factor [99]. Lesser expression of CD44v6 can slower down the cancer cell proliferation. 

#### 3.3.2. Breast

Signaling between HDAC5 and lysine-specific demethylase 1 (LSD1) facilitates the abnormal growth of breast cancer cell lines [100]. SFN treatment blocks the upstream transcription factor 1 (USF1) of HDAC5 inhibits the breast cancer cell lines. SFN with LSD1 inhibitor selectively inhibit the growth of cancer cell [101]. Withaferin A (WA), a metabolite derived from *Withania somnifera* and SFN promote breast cancer cell death by decreasing the BCL-2 and increasing the Bax expression [102]. Breast cancer cell line study showed that inhibition of Her2 and EGFR1 was found in the SFN treatment [103]. Activation of MAP kinases like ERK, JNK and p38 was observed in SFN treatment. Higher expression of MAPK signaling cascades induced the cyclin D1 and p21^CIP1^ gene which cause the cell cycle arrest at G1 phase [104]. Cyclin D1 is directly involved in carcinogenesis and also required for G1 phase of the cell cycle [105]. CDK-interacting protein 1 is associated with the process involved with the DNA damage to cell cycle arrest.

#### 3.3.3. Colon

SFN acetylate the DNA repair protein can able to selectively induce DNA damage of colon cancer cell [106]. Induction of apoptosis is associated with the higher expression of proapoptotic protein bax, activation of detoxifying enzymes, a release of cytochrome c and stimulate proteolysis of PARP [107]. Phase II enzymes induction stimulate the phosphorylation of ERK1/2 and Akt kinases [108]. Colon cancer cells growth arrest was more efficient in SFN with 3,3’-diindolylmethane [109]. Martin et al. (2018) reported that SFN treatment to the colon cancer rat model showed the down regulation of miR21, histone deacetylase inhibitors HDAC and human telomerase reverse transcriptase (hTERT) [110]. Polyposis in rat colon (Pirc) model rat injected with the SFN has higher level of DNA damage as well as loss of significant DNA repair regulator. It leads to cell cycle arrest and apoptosis. Higher C-terminal binding protein (CtBP) interacting protein (CtIP) alters the HAT/HDAC activity and its acetylation. Reduction in the expression of HDAC3, P300/CBP-associated factor (PCAF) and lysine acetyltransferase 2A (KAT2A/GCN5) weaken the repair mechanism of colon cancer cells. It is mainly associated with the homologous recombination (HR)/non-homologous end joining (NHEJ) activities [111].

#### 3.3.4. Prostate

SFN disrupts the microtubules of human prostate cancer cells. Cell death of DU145 and PC3 cell showed higher phosphorylation of ERK1/2 and caspase 3 and downregulation of α-tubulin [112]. Combination of SFN with the sorafenib and 5-fluorouracil drugs abolishes the pancreatic cancer stem cell characteristics. Inhibited pancreatic cancer cells found with lower expression of NF-κB [113]. Downregulation of inhibitors of apoptosis (IAP) family proteins and apoptotic protease activating factor 1 (Apaf-1) in SFN treatment on prostate cancer cell lines stimulated the cell death [114]. Mitochondrial biogenesis and DNA fragmentation of prostate cancer cells have lesser HIF1α in cancer cells. Selective regulation of SFN protects the damage of normal tissues. SFN plays an important role in the stabilization of Nrf2 and peroxisome proliferator-activated receptor-γ co-activator-1α (PGC1α) [58]. Nrf2 serves as a defense mechanism against oxidative stress and electrophilic toxicants by inducing more than a hundred of cytoprotective proteins, including antioxidants and phase II detoxifying enzymes [115]. Previously, Myzak et al., (2006) reported that prostate epithelial cells with SFN inhibit HDAC independent of Nrf2 [116]. SFN treatment demthylate the promoter region of Nrf2 which enhances its expression in prostate cancer [117]. Higher expression of Nrf2 reduces the cancer cell proliferation [118]. Detailed function of Nrf2 will be discussed in Section 4. Watson et al. (2014) reported that higher ubiquitination and acetylation in C-terminal nuclear localization signal peptide motif [119]. It is correlated with the dissociation from chromatin and decrease in global trimethyl-histone H4 is observed in many cancers including prostate cancer [120].

#### 3.3.5. Skin

In the prevention of squamous cell carcinoma yes-associated *protein* 1 (YAP1) and ∆Np63α are an important target of SFN. Tumor cell growth inhibition was highly correlated with the downregulation of YAP1 and ∆Np63α [121]. YAP1 is involved with the cell proliferation and suppressing apoptotic genes. ΔNp63α is the predominant p63 protein isoform. Other potential targets of SFN are caspase-1, interleukin-1β, NLR family apoptosis inhibitory protein 5/NLR family caspase-1 recruitment domain-containing protein 4 (NAIP5/NLRC4) and melanoma 2 (AIM2) inflammasome receptors [122]. For skin cancer, phase II enzymes activation was correlated with the transcription factors like Nrf2. Metastasis inhibition of SFN was evident with the prevention of epithelial-to-mesenchymal transition [123]. Higher expression of zinc finger E-Box binding homeobox (ZEB1) is associated with invasion and metastasis [123]. Prevention of tumor growth was related with suppression of COX-2/MMP2, ZEB1 and zinc finger *protein* SNAI1 (SNAI1) [48,123]. COX directly involved with mammary carcinogenesis [45]. MMP-2 has been linked to invasion and metastasis [50]. Figure 3 describes the strategic mechanism(s) involved by SFN to prevent tumor growth.

## 4. ROS-Nrf2-ARE Mediated Pathways in Downregulation of NF-κB

During the abnormal condition excessive generation of ROS creates imbalance in the cellular homeostasis. Induction of antioxidant enzymes scavenges the free ROS and helps to maintain the equilibrium. Meanwhile, as an immune response during abnormal conditions rapidly translocate Nrf2 to the nucleus from Nrf2-Keap1 complex. Potential ARE binding property of Nrf2 stimulate major antioxidant enzymes to detoxify the ROS [17,18]. Consequently, higher abundance of Nrf2 inhibits the NF-κB activation under tumor condition [124]. Heme oxygenase (HO-1) and glutamate-cysteine oxygenase (GCL) and its modifier unit (GCLM) regulate the Nrf2-Keap1 complex. BITC, PEITC and SFN have a potential to induce Nrf2 [125]. Consequently, higher abundance of Nrf2-driven antioxidant genes including HO-1, GCL, GCLM, GSH *S*-transferase P1, NAD(P)H:quinone oxidoreductase 1 (NQO1), γ-glutamylcysteine synthetase (γGCS), will be witnessed [18,125,126,127]. Oxidized low-density lipoprotein (oxLDL) is recognized to be related with prostate cancer, gastroenterological cancer, cardiovascular diseases [125,128,129]. oxLDL enhances intracellular oxidative stress activate translocation of NF-κB into the nucleus. Higher binding rate of NF-κB to the promoters of target genes triggers the progression of atherosclerosis. BITC, PEITC and SFN reduce the promoter activity of oxLDL to the NF-κB is mostly associated with the Nrf2-depenedent HO-1, GCLC and GCLM upregulation. Protection of BITC, PEITC and SFN against oxLDL-induced monocyte adhesion to the endothelium is likely associated with the Nrf2-dependent upregulation of the expression of HO-1, GCLC and GCLM [125]. Induction of phase II such as HO-1, γGCS and NQO1 on cultured fibroblasts (NIH3T3 cells) with ITCs treatment is correlated with the increase in phosphorylation of ERK1/2 (upstream binding region of Nrf2) [127].

Stimulation of Nrf2 promotes the antioxidant and phase II enzymes [18]. Following to it, in response to stress autophagy usually activated for the renovation of cell [130]. Heat shock and phase 2 response was controlled by heat shock factor 1 (HSF1) and Nrf2, respectively. In cutaneous squamous cell carcinoma caused by chronic exposure to UV radiation in SKH-1 hairless mice showed that *ortho*-hydroxylated double Michael acceptor *bis*(2-hydocybenzylidene)acetone (HBB2) inhibits the tumor formation. HBB2 induced the autophagy by mediating the Nrf2 and HSF1. However, both have opposite mechanism in suppressing tumor growth. Increase in concentration of HBB2 significantly reduced the expression of HSF1. Contrastingly, Nrf2 expression was increase in dose-dependent manner. Activation of autophagy by HBB2 was mediated with the Nrf2 [131]. Interestingly, Nrf2 activation stimulates viral replication in cancer cells and disrupts the type I interferon response via increased autophagy [132]. Hsf1 has also been reported to be involved with autophagy stimulated with the treatment of chemotherapeutic agents [133]. Anti-proliferative and induced apoptosis against cancer cell lines under ITCs treatments were occurred by both direct and indirect interactions. Especially, induction of Nrf2 attenuated the NF-κB indicate that ITCs mediated pathways involved with the ROS generation, antioxidant and other stress mechanisms of immune response. NF-κB is a redox sensitive complex and can be inhibited by higher antioxidants. Suppression of NF-κB activation is considered as a central mechanism in the prevention of tumor growth [24]. Majorly BITC, PEITC and SFN involved in the Nrf2-ROS-ARE dependent mechanisms to attenuate the NF-κB activation.

## 5. Direct and Indirect Inhibition of NF-κB by ITCs

Electrophilic characteristics of BITC, PEITC and SFN can able to regulate the NF-κB [125]. Binding of β-catenin and p65 on the NF-κB binding site was enhanced under 5 µM BITC and β-catenin binding on the TBE0 site was suppressed by treatment of 2.5 and 5 µM BITC. These results suggest that the interference of β-catenin binding on the TBE0 site by p65 is involved in the suppression of cyclin D1 gene expression [134]. BITC significantly enhanced the nuclear translocation and increased the sensitivity of NF-κB [83]. Further, Liu and Dey (2010) reported that PEITC inhibits NF-κB in a concentration-dependent manner. Inflammation and proliferation of colon cancer was attenuated at 10 μM PEITC. Suppression in the NF-κB expression was correlated with the histone modification. Similarly, in HT29 cell lines, PEITC induction of cytotoxicity and prevention of migration was strongly associated with the suppression of NF-κB along with MMP-9, PKS, MAPK, PI3K and AKT [81]. In human lung cancer cell A549, both BITC and SFN showed antiproliferation effect by binding to the tubulin whereas in colorectal cancer cell only BITC induced the nuclear translocation of p65. This might be due to the higher hydrophobicity of BITC than the SFN due to the presence of the aromatic ring. To regulate the NF-κB, structure of ITCs plays a vital role [134,135,136]. BITC treatment prevents the NF-κB in two ways. One is indirect inhibition of HDAC1 and HDAC3 [137] and another one directly block the phosphorylation of Ser-536 [138]. SFN selectively reduced DNA binding neither of NF-κB nor with nuclear translocation of NF-κB. SFN could either directly inactivate NF-κB subunits by binding to essential Cys residues or interact with glutathione or other redox regulators like thioredoxin and redox-factor 1 (Ref-1) relevant for NF-κB function. SFN can interact with thiol groups by formation of dithiocarbamate. Hence, SFN may impair the redox-sensitive DNA binding and transactivation of NF-κB [139]. In colorectal cancer cells, BITC treatment increased the interaction between p65 of NF-κB and β-catechin to block binding with positive cis element of cyclin D1 promoters. This process inhibits cyclin D1 expression and cell-to-cell proliferation [140]. Kim et al. (2014) reported that SFN inhibits the TPA-induced NF-κB activation and COX-2 expression by blocking both ERK1/2-IKKα and NAK-IKKβ complex [141]. Treatment of SFN inhibits the nuclear translocation of IκBα and p65 in NF-κB. Lesser expression of NF-κB reduces the inflammation of endothelial cell [142]. Self-renewal potential and tissue differentiation were eliminated by the blocking the TRAIL-induced NF-κB binding, CXCR4, Jagged1 (JAG1), Notch1, SRY (sex determining region Y)-box 2 (SOX2), Nanog and aldehyde dehydrogenase 1 (ALDH1), proteins involved in the cell cycle signaling [143]. Higher expression of COX-2 helps to progress and recurrence of bladder cancer cells. Suppression of COX-2 in SFN treatment reduced the binding of NF-κB to the COX-2 promoter region [144]. Schematic representation on regulation of ITCs to control the tumor growth via NF-κB has been demonstrated in Figure 4.

As enormous generation of ROS initiates and progress the cancer cell proliferation, it also induces the fibrosis [145]. Cellular fibrosis is associated with the proliferation of cancer [146]. Formations of excessive amount of fibrous connective tissues are generally termed as fibrosis. Cancer like uncertain circumstances and other chronic inflammation promotes various fibrosis such as pulmonary fibrosis in lung, cirrhosis in liver, atrial fibrosis and endo myocardial fibrosis in heart, glial scar in brain, skin fibrosis, kidney, intestine, bone marrow and so forth. As it is tissue specific yet it is difficult to control the fibrotic diseases [147]. Inhibition of ROS and other free radicals is a logical therapeutic intervention for fibrosis [145]. Fibrosis cause irreversible damage to the cell. Milczarek et al. (2011) reported that SFN treatment along with the 5-fluorouracil on Chinese hamster fibroblast cells line V79 prevented the antagonistic effect of 5-fluorouracil to the normal cells [148]. Anti-fibrotic potential of ITCs could attribute with its higher antioxidant and anti-inflammatory property on carbon tetrachloride (CCl_4_) induced fibrosis on male albino rats [149]. Recently, Kim et al. 2018 reported that in Sprague-Dawley rat induced fibrosis with CCl_4_ has lesser hepatic fibrosis upon AITC treatment. Activation on monocytes by AITC reduces the hepatic fibrosis in rats which are exposed repetitively to the CCl_4_ [150]. ITCs present in the cruciferous vegetables possess anti-inflammatory effects via reduction in the NF-kB activation, COX-2, TNF-alpha, IL-6 also controls fibrosis [149]. Even the cancer survivors are chronically plagued with fibrosis-related symptoms [146]. Radiation and chemotherapy are the potential cause for the fibrosis and its-related disease [146,147]. Therefore, more studies on the prevention of fibrotic disease by ITCs are required to control the fibrosis as well as prevent intervention during the treatment for cancer.

## 6. Conclusions

Cruciferous vegetables are receiving much attention due to its health-promoting properties and the successful clinical trials against cancer risk. Consumption of crucifers either as fresh or raw extract has more beneficial effects than the cooked or boiled as bioavailability of ITCs in former is higher than the later. BITC, PEITC and SFN target the proteins related with cell proliferation and homeostasis. Interaction with proteins involved in the repair of DNA damage, stimulate cell cycle arrest and induce the programmed cell death are the primary metabolism of ITCs in the prevention of tumor growth. NF-κB is considered as a major factor for cancer cell migration. BITC, PEITC and SFN are potentially interacting with the NF-κB and its regulatory pathway to prevent cancer cell growth. Further structural and functional genomics studies on the molecular interaction of ITCs with important signaling regulators of cancer cells could be very much useful for efficient diagnosis.

## Figures and Tables

**Figure 1 molecules-23-02983-f001:**
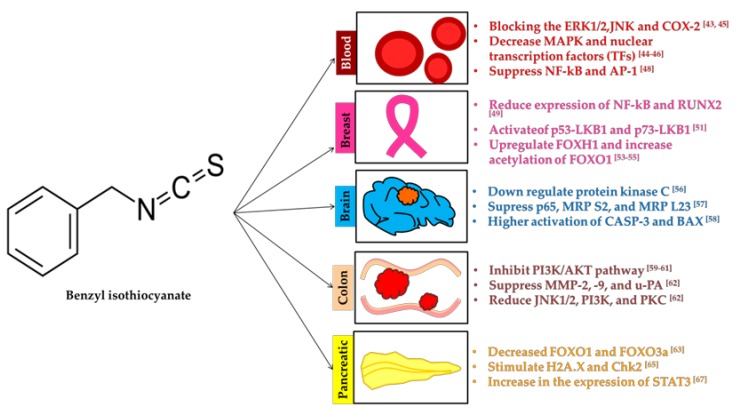
Key mechanisms involved by benzyl isothiocyanate (BITC) in the prevention of cancers.

**Figure 2 molecules-23-02983-f002:**
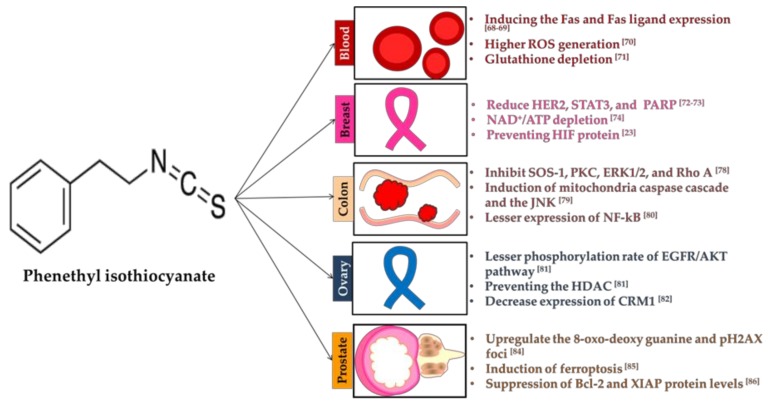
Important metabolic pathways involved by phenethyl isothiocyanate (PEITC) in the prevention of cancers.

**Figure 3 molecules-23-02983-f003:**
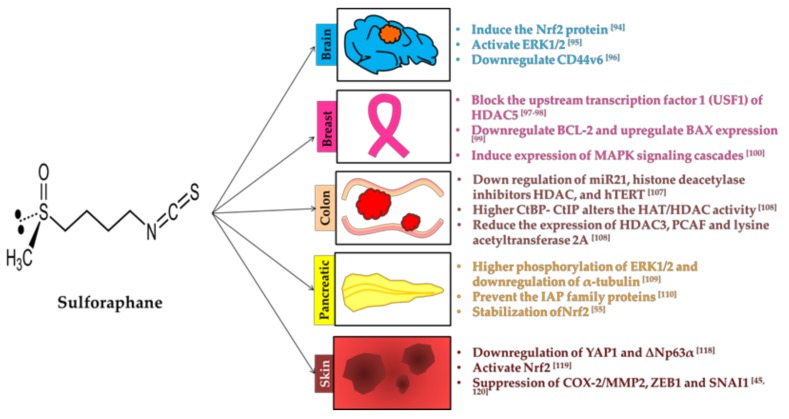
Vital process involved by sulforaphane (SFN) in the prevention of cancers.

**Figure 4 molecules-23-02983-f004:**
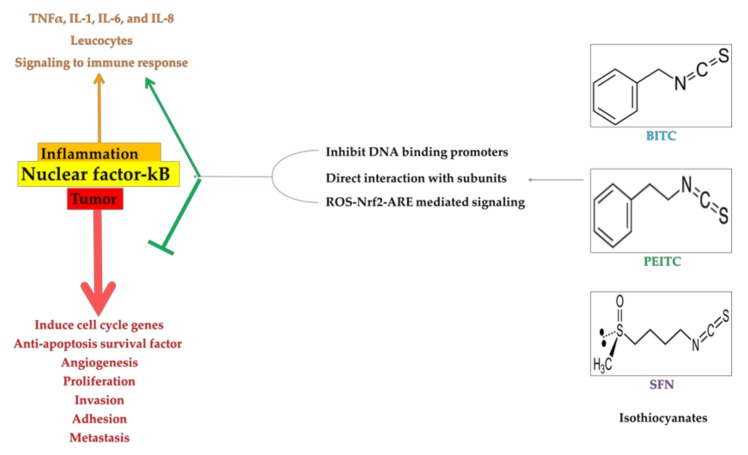
Role of Nuclear factor-kappaB (NF-κB) in the inflammation and regulation of isothiocyanates (ITCs) to control the tumor growth via NF-κB.

**Table 1 molecules-23-02983-t001:** Information about dietary sources of isothiocyanates and their precursors.

Isothiocyanate	Benzyl Isothiocyanate (BITC)	Phenethyl Isothiocyanate (PEITC)	Sulforaphane (SFN)
**Dietary Source**	Cabbage, garden cress, Indian cress	Watercress/turnip	Broccoli, Brussels sprouts, cabbage
**Precursor**	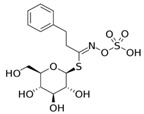 Glucotropaeolin	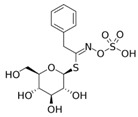 Gluconasturtiin	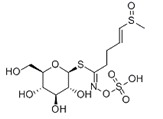 Glucoraphanin
**Structure**	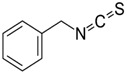	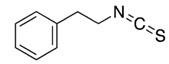	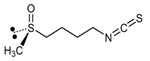

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
