# Peer review of "Anti-Carcinogenic Glucosinolates in Cruciferous Vegetables and Their Antagonistic Effects on Prevention of Cancers"

_molecules, 2018, doi:10.3390/molecules23112983_

Round 1
Reviewer 1 Report
The review article entitled “Anti-carcinogenic Glucosinolates in Cruciferous Vegetables and their Antagonistic Effects against Cancers” by Soundararajan and Kim describes the effects of glucosinolates (GSL) on a number of cancers. Authors suggest that antagonistic activity of BITC, PEITC, and SFN against cancer was due to interaction with NF-κB protein. The review also describes the chemo preventive effects of ITCs on cancer. It is suggested that glucosinolates can be used to prevent cancer. In some part of the review language needs little clarity. Authors missed to reference some of the published work, which is understand able.
Author Response
Reviewer 1:
Thank you for your comments. The manuscript has been revised according to your suggestions.
Comments and Response:
Point 1: In some part of the review language needs little clarity.
Manuscript language has been revised completely.
Point 2: Authors missed to reference some of the published work, which is understandable.
New references have been included in the revised version of manuscript.
Reviewer 2 Report
This is an interesting article reviewing the anticancer effects of 3 isothiocyanates (ITCs), BITC, PEITC and SFN. The authors first reviewed the effects of BITC, PEITC, and SFN towards various cancers specifically. The authors then reviewed the inhibition effects of these ITCs against NF-kB. However, the overall score of the presented version of manuscript did not merit publication in Molecules.
Major comments:
It is good that the authors discussed NF-kB as an specific target for ITCs in section 4. However, this section looks separated from the rest of the manuscript. Several targets (e.g. Nrf2) discussed in previous sections are connected with NF-kB pathway therefore these information are represented redundantly. The contents of section 4.1 and 4.2 are all background information that is unnecessarily long and therefore should be revised concisely. Besides, the author emphasized the double edged role of NF-kB, however did not fully discussed how the interaction of ITSc and NF-kB may or may not influence cancer cells through immune responses. Therefore, it is confusing to get the purpose of discussing NF-kB separately. Reconstruction of the manuscript structure or heavily revision/enrichment in section 4 is then recommended.
Some new researches should also be incorporated into this review. The reference list seems out-of-date.
There are several mixed styles and citations errors in the manuscript. The authors should carefully go through the manuscript again. Examples of these errors are listed below:
Line 416-419. 'BITC significantly ...'. It is clear that the research by Liu and Dey et al. is cited here, however no citation number is provided for the sentence.
There are mixed usage of numeric and linguistic expression of concentration. e.g. '2.5 or five uM'.
There are also some obvious citation style error. e.g. Line 399. 'Guttridge et al., [1999] suggested ...'.
Author Response
Thank you for your comments. The manuscript has been revised according to your suggestions.
Comments and Response:
Point 1: It is good that the authors discussed NF-kB as an specific target for ITCs in section 4. However, this section looks separated from the rest of the manuscript. Several targets (e.g. Nrf2) discussed in previous sections are connected with NF-kB pathway therefore these information are represented redundantly. The contents of section 4.1 and 4.2 are all background information that is unnecessarily long and therefore should be revised concisely. Besides, the author emphasized the double edged role of NF-kB, however did not fully discussed how the interaction of ITSc and NF-kB may or may not influence cancer cells through immune responses. Therefore, it is confusing to get the purpose of discussing NF-kB separately. Reconstruction of the manuscript structure or heavily revision/enrichment in section 4 is then recommended.
In the revised version of manuscript, Nrf2 and its mechanism on controlling NF-κB have been discussed in Section 4 as ROS-Nrf2-ARE pathway in NF-κB downregulation.
Background information and redundancy in the Section 5 (Direct and indirect inhibition of NF-κB by ITCs) has been precisely enriched.
ITCs role on the interaction with NF-κB through immune response has been discussed in section 4.
Overall, section 4 in the previous version of manuscript have been completely modified and separately discussed as section 4 and section 5
Point 2: Some new researches should also be incorporated into this review. The reference list seems out-of-date.
New references have been included in the revised version of manuscript.
Point 3: Line 416-419. 'BITC significantly ...'. It is clear that the research by Liu and Dey et al. is cited here, however no citation number is provided for the sentence.
Line 464-467: Liu and Dey, 2010 has been cited.
Point 4: There are mixed usage of numeric and linguistic expression of concentration. e.g. '2.5 or five uM'.
Point 5: There are also some obvious citation style error. e.g. Line 399. 'Guttridge et al., [1999] suggested ...'.
Both sentences have been removed in the revised version of manuscript.
Reviewer 3 Report
The paper is interesting.
I have only few comments regarding the literature citation
I suggest the Authors to add these papers in the introduction:
1. Popolo A, Pinto A, Daglia M, Nabavi SF, Farooqi AA, Rastrelli L. Two likely
targets for the anti-cancer effect of indole derivatives from cruciferous
vegetables: PI3K/Akt/mTOR signalling pathway and the aryl hydrocarbon receptor.
Semin Cancer Biol. 2017 Oct;46:132-137.
2: Tafrihi M, Nakhaei Sistani R. E-Cadherin/β-Catenin Complex: A Target for
Anticancer and Antimetastasis Plants/Plant-derived Compounds. Nutr Cancer. 2017
Jul;69(5):702-722.
3. Capasso R, Aviello G, Romano B, Borrelli F, De Petrocellis L, Di Marzo V, Izzo
AA. Modulation of mouse gastrointestinal motility by allyl isothiocyanate, a
constituent of cruciferous vegetables (Brassicaceae): evidence for
TRPA1-independent effects. Br J Pharmacol. 2012 Mar;165(6):1966-1977
Author Response
Reviewer 3:
Thank you very much for your valuable comments. The manuscript has been revision according to your suggestions.
Point 1: I suggest the Authors to add these papers in the introduction:
1. Popolo A, Pinto A, Daglia M, Nabavi SF, Farooqi AA, Rastrelli L. Two likely
targets for the anti-cancer effect of indole derivatives from cruciferous
vegetables: PI3K/Akt/mTOR signalling pathway and the aryl hydrocarbon receptor.
Semin Cancer Biol. 2017 Oct;46:132-137.
2: Tafrihi M, Nakhaei Sistani R. E-Cadherin/β-Catenin Complex: A Target for
Anticancer and Antimetastasis Plants/Plant-derived Compounds. Nutr Cancer. 2017
Jul;69(5):702-722.
3. Capasso R, Aviello G, Romano B, Borrelli F, De Petrocellis L, Di Marzo V, Izzo
AA. Modulation of mouse gastrointestinal motility by allyl isothiocyanate, a
constituent of cruciferous vegetables (Brassicaceae): evidence for
TRPA1-independent effects. Br J Pharmacol. 2012 Mar;165(6):1966-1977
Above recommended references have been cited in the revised version of manuscript
Popolo et al., 2017 has been cited in Line 44-49 and 148-150.
Tafrihi et al., 2017 has been cited in Line 41.
Capasso et al., has been cited in Line 73-75.
Reviewer 4 Report
The present manuscript by Soundararajan and Kim is a review on “anti-carcinogenic glucosinolates” and their “antagonist effect in cancer”.
The term “prevention of cancer” should be present in the title to avoid misunderstanding with cancer treatment.
The originality of the present manuscript, related to other reviews on glucosinolates, is to highlight NF-kB as a “potential target” for glucosinolates. However, the term “target” is over-interpreted. It should rather be considered as a deregulated protein (indirect effect).
The implication of NRF2 is overall much described in the literature and the impact of glucosinolates on NFkB pathway may depends more likely on NRF2 transcription factor. This should be strongly discussed.
The role of NFkB should be more explicitly described relatively to NRF2 prior to envisage a publication in Molecules. Otherwise, the manuscript should encompass other deregulated transcription factors
Similarly, the authors focused on the inflammatory process but should also associate informations related to the importance of fibrosis in cancer development and the prevention of fibrosis.
Figures 1 to 3 should be combined as a single table and corresponding references numbers added.
Some recent references are missing and should be discussed:
Mohammed et al, 2017 ; Lee et al, 2017 ; Sturm and Wagner 2017; Kaczmarek et al., 2018; Curran et al., 2018.
This is also the case for the recent and interesting review Arumugam et al., 2018 that present many points common to the present review in terms of cellular processes affected by glucosinolates and related to cancer.
Reference Dayalan Naidu et al., 2018 should also be included and discussed for HSF1, as reference and Guzman-Perez et al 2016 for FOXO1 modulation.
Typing errors, among others :
Lane 44 : B. rapa (Br) to Brassica rapa (Br)
Lane 411 >ten µM to >10 µM
Author Response
Thank you very much for your valuable comments. The manuscript has been revision according to your suggestions.
Point 1: The term “prevention of cancer” should be present in the title to avoid misunderstanding with cancer treatment.
“Prevention of Cancer” has been included in the topic.
Point 2: The implication of NRF2 is overall much described in the literature and the impact of glucosinolates on NFkB pathway may depends more likely on NRF2 transcription factor. This should be strongly discussed.
Role of Nrf2 has been discussed strongly in section “ROS-Nrf2-ARE pathway in NF-κB downregulation”.
Point 3: The role of NFkB should be more explicitly described relatively to NRF2 prior to envisage a publication in Molecules. Otherwise, the manuscript should encompass other deregulated transcription factors
Interaction of Nrf2 with NF-κB has been discussed in section 4.
Point 4: Similarly, the authors focused on the inflammatory process but should also associate informations related to the importance of fibrosis in cancer development and the prevention of fibrosis.
Association of fibrosis with cancer and studied related to fibrosis and ITCs has been discussed in section 5 (paragraph 2, Line 491-510).
Point 5: Figures 1 to 3 should be combined as a single table and corresponding references numbers added.
As most of the review articles highlight the key metabolisms of GSLs in table format, we would like to give some pictorial representation. As per the reviewer suggestion, references have been cited accordingly in the figures 1-3.
Point 6: Some recent references are missing and should be discussed:
Mohammed et al, 2017 ; Lee et al, 2017 ; Sturm and Wagner 2017; Kaczmarek et al., 2018; Curran et al., 2018.
Above recommended references have been added in the revised manuscript.
Mohammed et al, 2017 – Line 502-508.
Lee et al, 2017 – Line 172-173
Sturm and Wagner 2017 – Line 87-92, 424-427, and 438.
Kaczmarek et al., 2018 – Line 152-154.
Curran et al., 2018 – Line 149-152.
Point 7: This is also the case for the recent and interesting review Arumugam et al., 2018 that present many points common to the present review in terms of cellular processes affected by glucosinolates and related to cancer.
During the preparation of this manuscript we haven’t gone through Arumugam et al., 2018. If any similarities in points that is probably only due to the common topic ie., glucosinolates and its effects on cancer inhibition.
Point 8: Reference Dayalan Naidu et al., 2018 should also be included and discussed for HSF1, as reference and Guzman-Perez et al 2016 for FOXO1 modulation.
Reference Dayalan Naidu et al., 2018 has been added for HSF1 (Line 439-446).
Reference Guzman-Perez et al 2016 has been added for FOXO1 (Line 201-203, and 287-390).
Point 9: Typing errors, among others :
Lane 44 : B. rapa (Br) to Brassica rapa (Br)
Line 51: B. rapa (Br) has been changed to Brassica rapa (Br)
Point 10: Lane 411 >ten µM to >10 µM
Sentence has been removed in the revised version of manuscript.
Round 2
Reviewer 2 Report
Thanks for the revision, the current version is acceptable for publication in Molecules.
Reviewer 4 Report
The corrected version of the manuscript is now ready for publication